# Legless soft robots capable of rapid, continuous, and steered jumping

Rui Chen [1,9 ✉], Zean Yuan [1,9], Jianglong Guo [2,9], Long Bai [1,9], Xinyu Zhu[1], Fuqiang Liu [3], Huayan Pu [4 ✉], Liming Xin [5], Yan Peng [6], Jun Luo[1,4], Li Wen [7] & Yu Sun [8]

Jumping is an important locomotion function to extend navigation range, overcome obstacles, and adapt to unstructured environments. In that sense, continuous jumping and direction adjustability can be essential properties for terrestrial robots with multimodal locomotion. However, only few soft jumping robots can achieve rapid continuous jumping and controlled turning locomotion for obstacle crossing. Here, we present an electrohydrostatically driven tethered legless soft jumping robot capable of rapid, continuous, and steered jumping based on a soft electrohydrostatic bending actuator. This 1.1 g and 6.5 cm tethered soft jumping robot is able to achieve a jumping height of 7.68 body heights and a continuous forward jumping speed of 6.01 body lengths per second. Combining two actuator units, it can achieve rapid turning with a speed of 138.4° per second. The robots are also demonstrated to be capable of skipping across a multitude of obstacles. This work provides a foundation for the application of electrohydrostatic actuation in soft robots for agile and fast multimodal locomotion.

[1] State Key Laboratory of Mechanical Transmissions, Chongqing University, Chongqing 400044, China. [2] School of Science, Harbin Institute of Technology (Shenzhen), Shenzhen 518055, China. [3] College of Mechanical and Vehicle Engineering, Chongqing University, Chongqing 400044, China. [4] School of Mechatronics Engineering and Automation, Shanghai University, Shanghai 200444, China. [5] School of Computer Engineering and Science, Shanghai University, Shanghai 200444, China. [6] Research Institute of Unmanned Surface Vessel Engineering, Shanghai University, Shanghai 200444, China. [7] School of Mechanical Engineering and Automation, Beihang University, Beijing 100191, China. [8] Department of Mechanical and Industrial Engineering, University of Toronto, Toronto, Canada. [9] These authors contributed equally: Rui Chen, Zean Yuan, Jianglong Guo, Long Bai. ✉email: cr@cqu.edu.cn; phygood_2001@shu.edu.cn

As an important locomotion function of terrestrial robots, jumping or leaping is useful for them to effectively extend their navigation range, overcome challenging obstacles, and enhance their adaptability in unstructured environments[1–3]. However, enhancing single-jump performance (jumping height (JH) and jumping distance (JD)) of soft jumping robots to improve their obstacle-crossing ability and accelerating jumping frequency to increase their navigation efficiency at the same time are two grand engineering challenges. Researchers have developed soft or partially soft jumping robots capable of forward navigations and driven by integrated springs[4–17], shape memory alloys (SMAs)[18–20], magnetic actuators[21], light-powered actuators[22,23], dielectric elastomer actuators (DEAs)[24–26], pneumatic actuators[27–29], chemical actuators[30–33], motors[34–36], and poly-vinylidene difluoride (PVDF) actuators[37]. Some of them, which are energy-storing jumping robots[4–25], have excellent single-jump performance, but always at the cost of navigation efficiency due to the necessity of the additional elastic energy-storage process. The prolongation of the energy-storage process increases the JH but decreases the landing stability and reduces the jumping frequency. On the other hand, soft jumping robots actuated by pneumatic actuators[27–29], chemical actuators[30–33], and motors[34–36] have been demonstrated but have complicated navigation strategies and structures. Lightweight soft hopping robots based on DEAs[26] and PVDF actuators[37] can simply jump by bending their body parts without additional energy-storing, which can lead to rapid jumping frequencies, but their JHs and JDs are not enough (<0.25 body height) to meet the requirements of crossing obstacles.

Hydraulically amplified self-healing electrostatic (HASEL)[38–41] actuators, which can achieve linear motion by electro-hydrostatically changing the distribution of internal liquids, have been demonstrated to achieve remarkable continuous actuation performance with actuation strains up to 118%, strain rates of about 7500% s$^{-1}$, and a peak specific power of 156 W/kg. The stacked quadrant donut HASELs presented by Mitchell et al.[40] can achieve continuous vertical jumping by rapidly changing internal liquid distribution with a JH of about 1.67 body heights. This electrohydraulic actuation method, which can generate the energy required for jumping in a very short time without the need for a complicated energy-storing process, is a potential solution for rapid obstacle-crossing robots. However, the diffusion-like isotropic liquid flow of a donut HASEL actuator cannot generate the energy of the forward jumping since the kinetic energy of the liquid in all directions was canceled out, which caused the loss and waste of energy. The jumping caused by the partial expansion of the liquid pouch can only keep the partial actuator off the ground and the backflow of the dielectric liquid relies solely on gravity, making it slow to return to the original state after landing (Supplementary Fig. 1a). Therefore, it is still challenging for HASEL jumpers to (1) achieve enhanced single-jump performance without stacking, (2) achieve rapid restoration, and (3) generate forward jumping and steered jumping.

Based on the flexible electrical-driven liquid redistribution method of HASEL series actuators[38–41], the actuator structure was redesigned to make liquid flow anisotropically, utilize the kinetic energy generated by liquid redistributions, and achieve forward jumping. In addition, using saddle-shaped bending based on an elastically deformable frame-membrane structure to achieve locomotion is common in DEAs[42], which inspired us to use rapid bending and rebound based on electrohydrostatic principle and frame to enhance the jumping performance of actuators. In this work, we propose an electrohydrostatically driven tethered legless soft jumping robot (LSJR) with rapid, continuous, steered jumping and obstacle-crossing capabilities based on a soft electrohydrostatic bending actuator (sEHBA). We

show that the characteristic of sEHBA's rapid response leads to a short actuation time (~10 ms). The LSJR can be used to achieve a JH of 7.68 body heights, a JD of 1.46 body lengths in a single jump, and a continuous forward jumping speed of 390.5 mm/s (6.01 body lengths per second) with a frequency of 4 Hz. We also demonstrate that the integration of two LSJRs can readily achieve rapid steered jumping. The turning speed of the dual-body LSJR was able to reach 138.4°/s$^{-1}$, which is the fastest among existing soft jumping robots. Furthermore, we show that LSJR's rapid continuous jumping locomotion can cross various obstacles, including slopes, wires, single steps, continuous steps, ring obstacles, gravel mounds, and cubes of different shapes, some of which are larger than the robot.

## Results

**Design concept and movement principle of the LSJR.** In order to use the anisotropic liquid flow to achieve forward jumping caused by unbalanced energy, we heat-sealed a HASEL like actuator into a semicircular separated HASEL (SCS-HASEL) actuator composed of two semicircular liquid pouches based on the zipping mechanism[43] and it showed better jumping performance (Supplementary Fig. 1b). Then, the dielectric liquid in the rear semicircular pouch of the SCS-HASEL actuator was replaced with an equal volume of air and the covered electrodes of the rear semicircular pouch were removed so that the dielectric liquid can flow anisotropically relative to the entire actuator. As expected, it can be found that the special liquid–air layout can make the liquid–air actuator jump forward even though the air pouch dragged on the ground (Supplementary Fig. 1c). This was because the electrodes squeezed the liquid dielectric to make it flow forward quickly, thus giving it the initial kinetic energy that can be used to provide forward kinetic energy to the liquid–air actuator. The detailed experimental results can be seen in Supplementary Movie 1.

By observing the three types of actuators above (Supplementary Fig. 1), it can be found that their jumps were all caused by the partial expansion of the liquid pouch. This kind of jump was unstable and unreliable, and the JH was insufficient. Furthermore, the edges of the SCS-HASEL actuator and liquid–air actuator were unrestricted so that every jump had randomness and the initial state cannot be completely restored. Fixing a predeformed frame on the edge is a well-known restriction method in DEAs[42] and has not been used for HASEL series actuators. Combining the frame to transform the linear motion of the electrohydraulic actuators into a saddle-shaped bending motion provided the possibility of an efficient type of jumping method. Therefore, the non-prebending ring frame and the prebending ring frame were combined to the SCS-HASEL actuator and the liquid–air actuator, and then their single-jump performance was tested, respectively (Supplementary Fig. 2). The left half of Supplementary Fig. 2 shows the moment (11.76 ms after applying voltage) the actuators took off from the ground. The prebending frames fixed on the edges of these actuators can guide the direction of deformation, making the deformation of the actuators with prebending frames more regular and closer to the saddle shape than the actuators with plane frames and achieve better jumping performance. The liquid–air actuator with a prebending frame, which was also the sEHBA we needed, can jump higher and further, and it can return to its original state immediately after landing. Clearly, using the frame with more regular deformation to touch the ground and generate jumping energy is a more stable and reliable jumping mechanism than relying on the partial expansion of the liquid pouch (Supplementary Movie 1).

In this paper, we propose an electrohydrostatically driven LSJR (Fig. 1) that has rapid, continuous, steered jumping and obstacle-

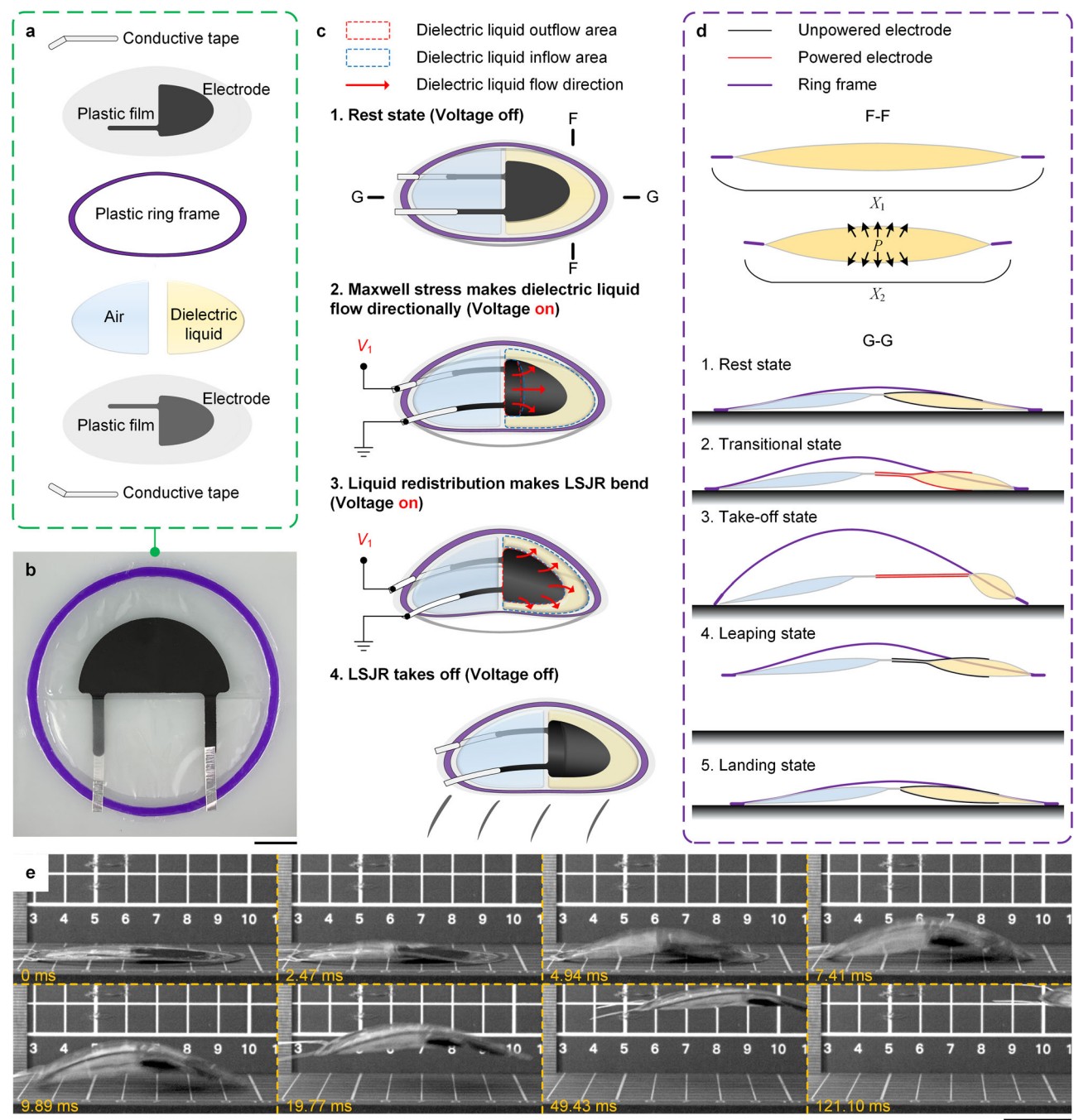

**Fig. 1 LSJR detailed design and motion principle. a** The LSJR consists of two plastic semicircular pouches printed with flexible electrodes. The front pouch is filled with a dielectric liquid, and the rear is filled with air with the same volume. A flexible plastic ring frame is fixed on the edge and is prestrained. Note that the rear air pouch functions to ensure that the pre-curved frame is consistent and maintains structural balance during the flight. **b** The LSJR prototype (1.1 g). Scale bar, 1 cm. **c** Schematic diagram of the LSJR jumping process. By the application of a high voltage to the two electrodes, the LSJR is energized to bend itself to generate forces and energy for forward jumping. During the voltage application, Maxwell stress squeezes the dielectric liquid and makes it flow laterally into the portion of the front pouch that is not covered by the electrodes (from the liquid outflow area to the liquid inflow area). **d** Cross-sectional views (e-e and f-f) of the LSJR: e-e denotes the deformation of the front pouch, whereas f-f shows the e-e deformation-driven whole-body bending and jumping. **e** Snapshots of the LSJR jumping, where 10 kV is applied to the actuator. Scale bar, 2 cm.

crossing capabilities based on the sEHBA. The special liquid–air layout and the semicircular zipping structure can be used to make the internal liquid flow anisotropically and rapidly to generate a great amount of forward kinetic energy, which is offset and wasted in HASEL actuators. Meanwhile, the prebending frame fixed on the edge of the sEHBA can be used to guide the deformation direction to achieve fast bending locomotion which

can be used to generate vertical and forward kinetic energy. The special liquid–air layout and prebending frame structure greatly enhance the jumping performance of electrohydraulic actuators and enable rapid, continuous, and steered jumping without stacking. The LSJR can jump simply by quickly liquid flowing and body bending, thereby greatly shortening the propulsive interval (~10 ms). The stored elastic energy associated with body bending

can help the robot quickly restore its original shape to avoid affecting the next jump. The detailed design iterations are demonstrated in Supplementary Fig. 3.

The LSJR consists of two flexible plastic semicircular pouches printed with flexible electrodes that were connected with two conductive tapes for potential electric wire connections (Fig. 1a). The two pouches were made of biaxially oriented polypropylene (BOPP) films. The front pouch was filled with a dielectric liquid and the rear was filled with the same volume of air. A flexible plastic polyvinyl chloride (PVC) ring frame was fixed on the edge and prestrained. Note that the rear air pouch, which is similar to the tail of animals and is used to maintain the balance of the jumping and landing posture, played an important role in the whole structure of the LSJR. To further enhance the jumping performance of LSJR, the air in this pouch can be replaced with helium or other less dense and non-explosive gas. The prototype of an LSJR (1.1 g) is shown in Fig. 1b. The detailed fabrication process can be seen in "Methods".

The LSJR is energized to bend itself to generate forces and energy for forward jumping. After the application of a high voltage to the electrodes, Maxwell forces attract the electrodes, squeezing the dielectric liquid from the outflow area to the inflow area with no electrode coverage (Fig. 1c). The rapid and anisotropic flow can generate a horizontal initial kinetic energy. The increased electrostatic force between the electrodes of the front pouch causes a rapid liquid flow, which increases the thickness of the cross-section F-F and decreases its lateral length from $X_1$ to $X_2$, as the pouch is inextensible. This deformation, plus the pre-strain of the ring frame to keep the bottom electrode from touching the substrate and facilitate the further bending of the frame during actuation (cross-section G-G), pulls the front and rear ends closer to each other (Fig. 1d). The instantaneous partial deformation of the frame results in instant bending of the overall frame, which propels the robot body into the air. Snapshots of a rapid (~10 ms) take-off process is presented in Fig. 1e. After take-off, the initial horizontal velocity of LSJR is determined by the horizontal ground reaction forces at the frame ends. The forces are caused by the moving dielectric liquid flow and frame bending. The initial vertical velocity of the LSJR is determined by the vertical ground reaction forces at the frame ends, which are caused by the frame bending. During the leaping state, the ring frame quickly releases its elastic energy, and the dielectric liquid flows back, restoring the robot to the original state in preparation for the next jump after landing. Note that the low-profile robot design makes both the jumping and landing stable with no capsizing. The detailed theoretical analysis of the locomotion mechanism is demonstrated in Supplementary Note 1.

**Single-jump characterization**. JD and JH are two important performance measures that can be used to characterize the jumping performance of the LSJR. If the same robot material and size are maintained, the electrode area/nonelectrode area ratio, magnitude of the voltage application, the mass of the load, and the ring frame prebending level are important parameters influencing the jumping performance of the LSJR. Figure 2a shows the untethered single-jump process and related parameters. The two aluminum electrodes connected to the tail of the LSJR were freely placed on two copper electrodes (top left inset in Fig. 2a), aiming to eliminate the influence of electric wires on the jumping performance. Note that (1) three LSJRs with the same parameters were fabricated, and the differences in the JD and JH results were all within 10%, and (2) for each jumping experiment, 10 repeated tests were performed in the same laboratory environment, and average and one standard deviation values of each 10 results were

reported. Furthermore, applying a voltage of the same polarity would cause the charge to be retained and to accumulate inside the actuators, which would prevent the actuators from fully returning to their initial position, thus affecting the results of the next experiment. Therefore, the polarity would be reversed and a waiting time of 60 s would be used to alleviate charge retention after each experiment (top right inset in Fig. 2a).

We define $r$ = electrode area:nonelectrode area (of the front pouch) and fabricated three robots with $r$ = 2:1, 1:1, and 1:2. Figure 2b, c shows that the robot with $r$ = 1:1 produced a larger JD and JH. At a low voltage (0~3 kV), the robot deformed slowly, and the deformation force was not large enough for jumping. Average values of JD = 95.0 mm (1.46 body lengths) and JH = 30.7 mm (7.68 body heights) were achieved when the applied voltage was 10 kV and $r$ = 1:1. Carrying a 1 g load (0.91 body weight) decreased the JD (56.0 mm) and JH (20.0 mm) to 59% and 65% of the no-load condition, respectively, bringing a jumping performance reduction of 38%. Carrying a 2 g load (1.82 body weight) decreased the JD (33.7 mm) and JH (8.1 mm) to 35 and 26% of the no-load condition, resulting in a jumping performance reduction of 70%, which was nearly twice that of a 1 g load condition. We used different body heights (BH) to represent different prebending levels and fabricated three robots with BH = 2, 4, and 6 mm. Figure 2d, e shows that the robot with BH = 4 mm produced a larger JD and JH. In addition, the LSJR with 2 mm BH had a greater average relative jumping height (RJH) of 9.4 when 10 kV was applied, whereas the average RJHs of the LSJR (BH = 4 mm) and the LSJR (BH = 6 mm) were 7.7 and 4.2, respectively. Comparing the experiment results (Fig. 2b, c) of the LSJR ($r$ = 2:1, load = 0 g), the LSJR ($r$ = 1:1, load = 0 g), and the LSJR ($r$ = 1:2, load = 0 g), an apparent difference in their JHs but a very small difference in JDs can be found. According to Supplementary Note 1, the amount of dielectric liquid flow had a much greater impact on the initial horizontal velocity $v_x$ than on the initial vertical velocity $v_y$. The bigger ratio $r$, which can affect the size of the electrode coverage area, led to a bigger volume $\Delta V_{oil}$ of dielectric liquid flow, a faster velocity $v_{oil}$ of dielectric liquid flow, and the faster bending rate of the ring frame. It caused a bigger horizontal initial kinetic energy of the moving dielectric liquid flow and greater vertical ground reaction forces, leading to a bigger JD and JH. However, the excessive $r$ (e.g. $r$ = 2:1) not only affected the flexibility of BOPP films and hindered the normal bending of the frame, reducing the vertical ground reaction forces, but also led to a smaller $\Delta V_{oil}$ and a lower $v_{oil}$, reducing the horizontal initial kinetic energy of the moving dielectric liquid flow. Therefore, the ratio ($r$ = 1:1) was the most appropriate in this experiment.

**Continuous jumping on different substrates**. Rapid continuous forward jumping is a useful locomotion capability. Continuous forward jumping speed (CFJS) is an important performance feature characterizing a continuous forward jumping robot. If the same robot is used, the substrate surface roughness, actuation frequency, and magnitude of the applied voltage are important parameters influencing the jumping performance of the LSJR. Sufficient substrate surface roughness prevents the robot from slipping during continuous motions, and an appropriate actuation frequency enables a quicker continuous movement. Note that (1) for each jumping experiment, 10 repeated tests were performed in the same laboratory environment, and the obtained results were reported as the average and one standard deviation, and (2) the polarity was reversed to alleviate charge retention.

It is shown in Fig. 1e that applying 10 kV to the robot for 10 ms or more was a necessary condition for a successful jump. This resulted in a jumping time of almost 250 ms (Supplementary

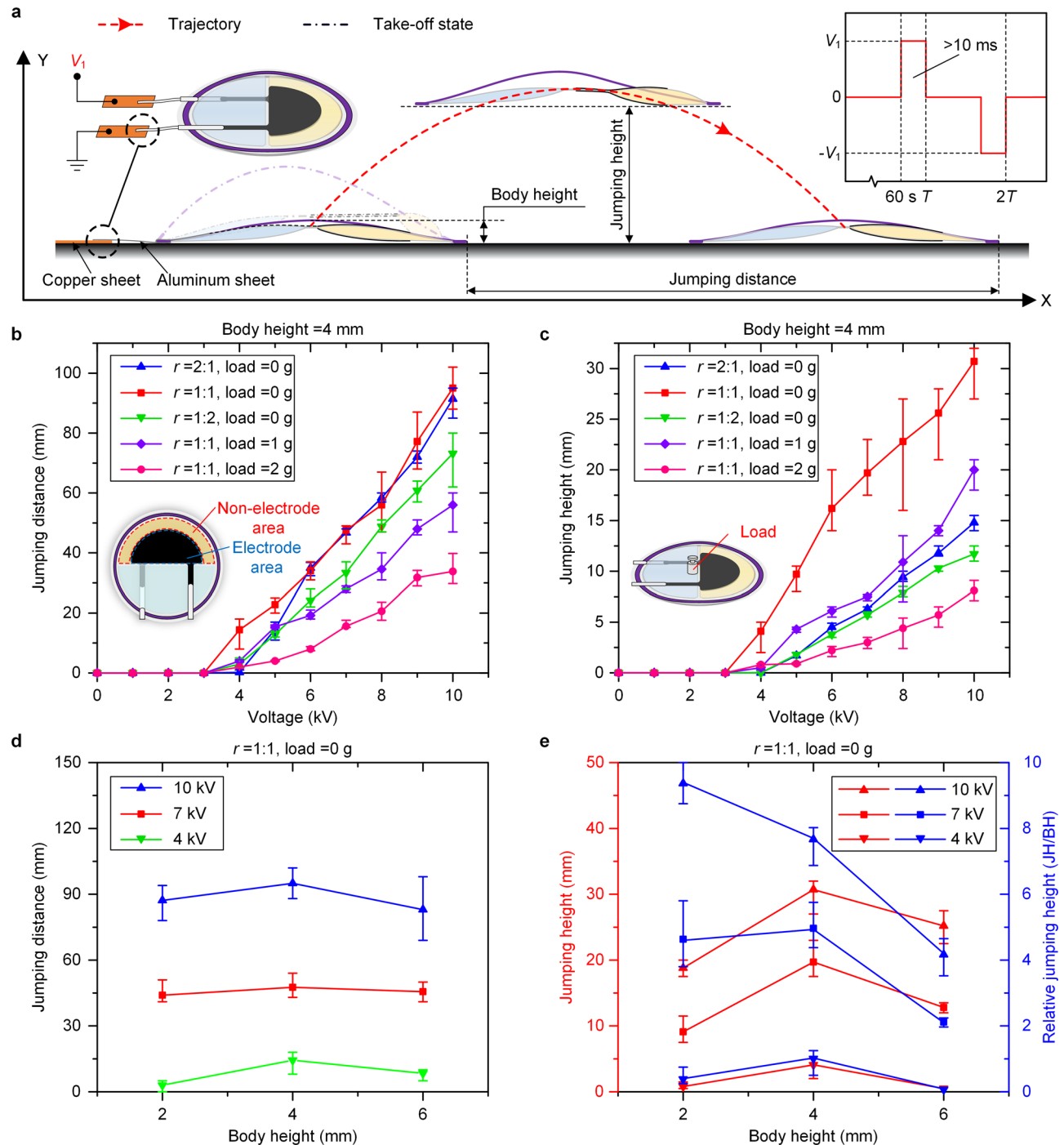

**Fig. 2 Single-jump characterization results.** See also Supplementary Movie 2. **a** Untethered single-jump process and parameters. The top left inset shows the electrical connections. The top right inset is the voltage application strategy in the experiment. **b** The relationship between JD and applied voltage under different loads (0, 1, and 2 g) and different electrode area/nonelectrode area ratios (2:1, 1:1, and 1:2). **c** The relationship between JH and applied voltage under different loads (0, 1, and 2 g) and different electrode area/nonelectrode area ratios (2:1, 1:1, and 1:2). **d** The relationship between JD and applied voltage at different body heights (2, 4, and 6 mm). **e** The relationship between JH, RJH, and applied voltage at different body heights (2, 4, and 6 mm).

Movie 2) and a typical actuation frequency of 4 Hz. Therefore, we set the power-on time to 10 ms and selected a series of test frequencies, i.e., 8, 4, 2, 1, 0.5, and 0.25 Hz, to test the influence of actuation frequency. In the experiments, the LSJR moved for four cycles at each frequency, and CFJS was calculated. In Supplementary Movie 3, we demonstrated continuous forward jumping locomotion of the soft robot on four substrates, i.e. a glass plate, a

paper plate, a PVC plate, and a wood plate, with different surface roughness (Supplementary Fig. 10). Figure 3a–d shows that at 4 Hz and 10 kV, the robot achieved a greater average CFJS = 390.5 mm/s (6.01 body lengths per second) on the wood substrate, whereas an average CFJS = 95.6 mm/s (1.47 body lengths per second) was achieved on the glass plate because it was difficult for the smooth glass plate to provide sufficient friction.

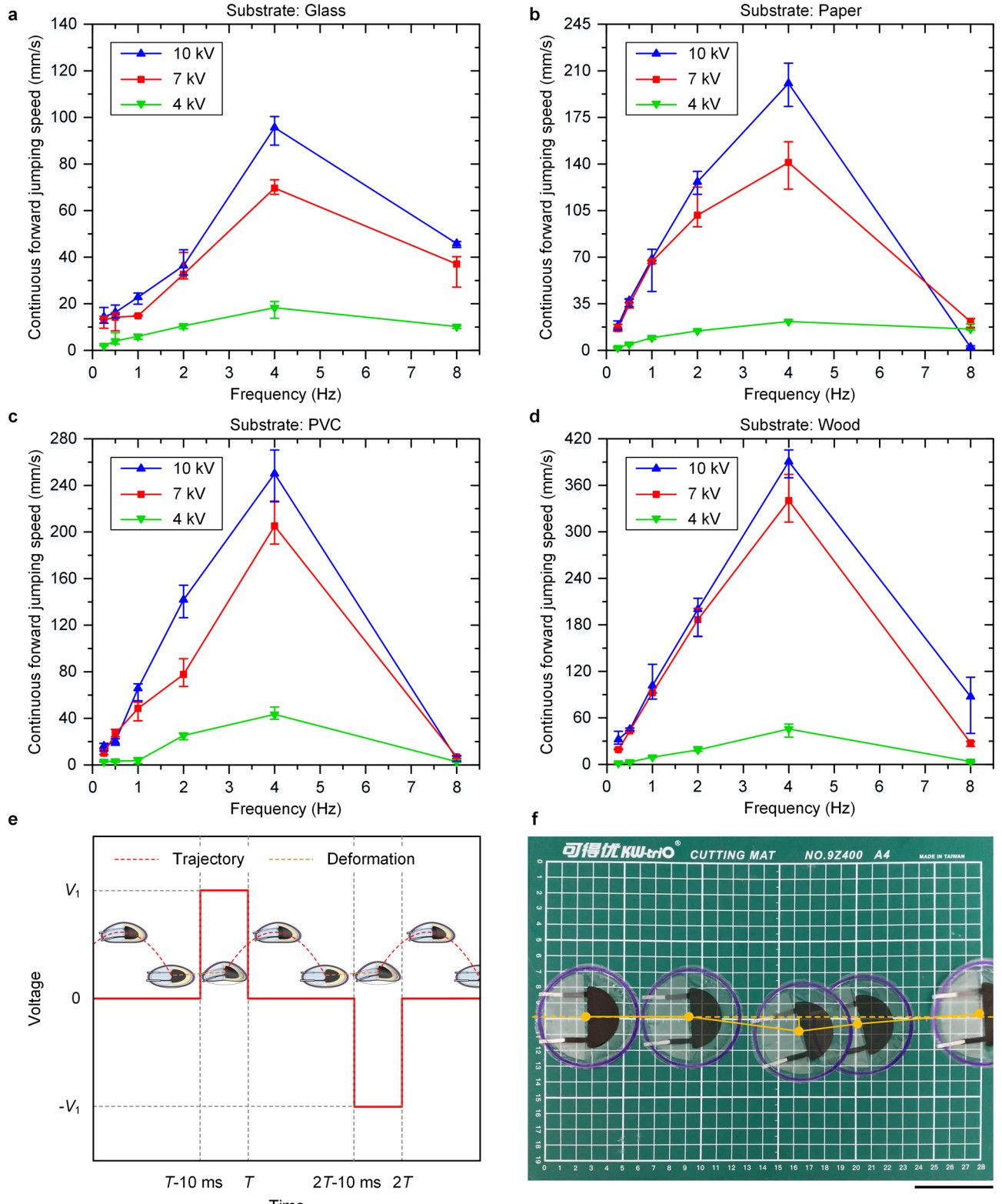

**Fig. 3 Continuous jumping on different substrates.** See also Supplementary Movie 3. **a** The relationship between CFJS and actuation frequency on the glass plate, where the slowest average CFJS = 95.6 mm/s (1.47 body lengths per second) was obtained at 4 Hz and 10 kV. **b** The relationship between CFJS and actuation frequency on the paper plate. **c** The relationship between CFJS and actuation frequency on the PVC plate. **d** The relationship between CFJS and actuation frequency on the wood plate, where the fastest average CFJS = 390.5 mm/s (6.01 body lengths per second) was obtained at 4 Hz and 10 kV. **e** The voltage application strategy and the corresponding robot motion states. **f** Composite image of the initial position and four landing points in continuous jumping on the PVC plate when CFJS = 250.1 mm/s at 4 Hz and 10 kV. The angle deviation of each jump was less than 8°. Scale bar, 5 cm.

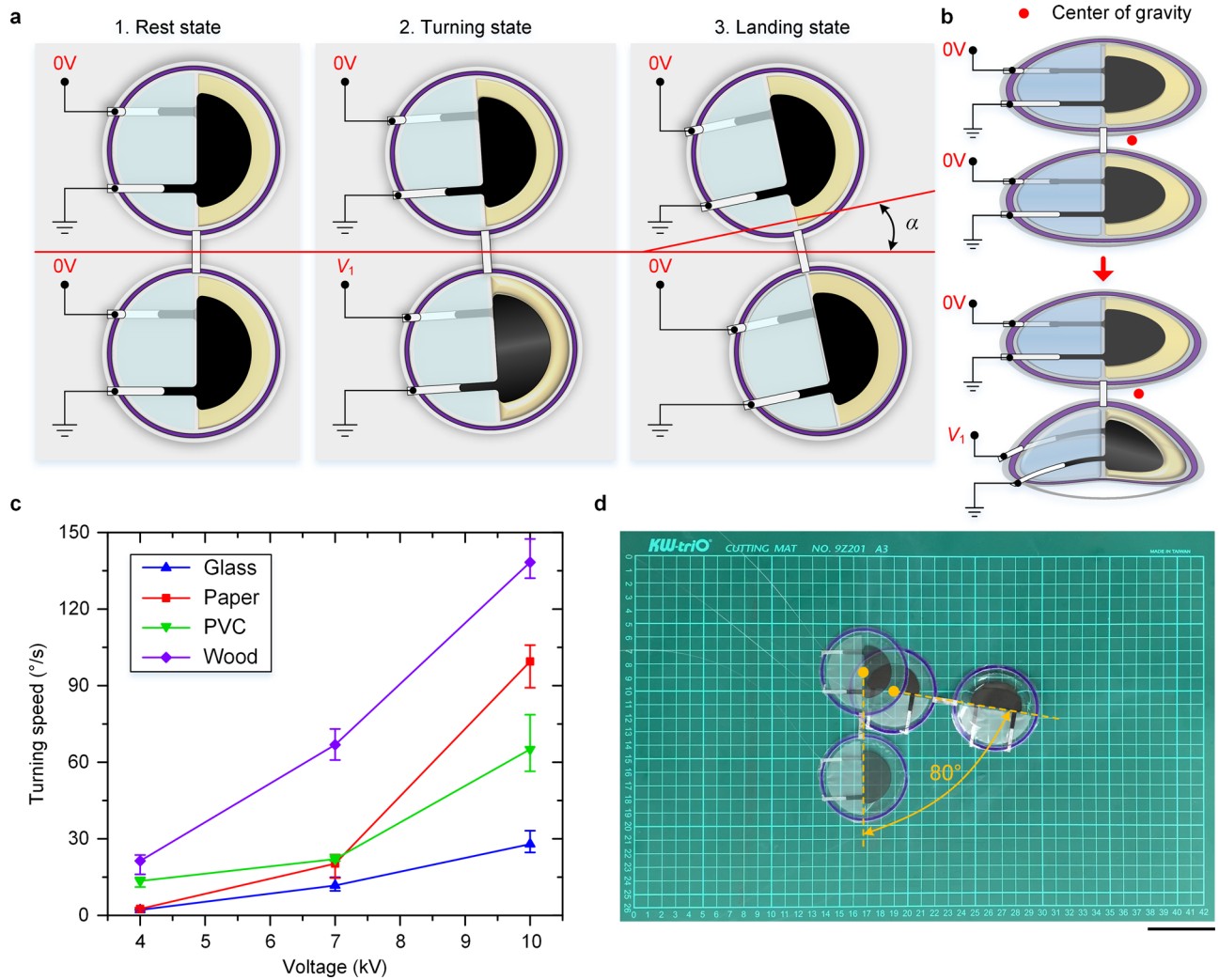

**Fig. 4 Turning results of the dual-body LSJR.** See also Supplementary Movie 4. **a** Schematic diagram of the dual-body LSJR turning process, which consists of the rest state, the turning state, and the landing state. Over each voltage cycle, the robot turns by an angle of $\alpha$. **b** Center of gravity of the dual-body LSJR. **c** The relationship between TS and applied voltage on the four substrates. **d** Composite image of the initial and final positions during a continuous turning procedure on the PVC plate with a speed of 65.0°/s. The robot took 1.23 s to turn 80° at 10 kV and 4 Hz. Scale bar, 5 cm.

As the actuation frequency increased, the CFJS increased. At a frequency higher than 8 Hz, after the LSJR took off and left the substrate, another actuation cycle started while the robot was still in the air. Each actuation cycle should only occur after the LSJR had landed smoothly; otherwise, the next jumping performance is impaired. Figure 3e shows the constant voltage application waveform and the corresponding robot states. Applying a voltage of the same polarity would cause the charge to be retained and to accumulate inside the actuators, which would prevent the actuators from fully returning to their initial position. Reversing the polarity mitigated charge retention within the actuator during continued cycling. The voltage application period was required to be long enough to prevent the LSJR from entering the next actuation cycle before landing. Therefore, changing the voltage waveform in real time through visual recognition and other means and making the LSJR enter the next actuation cycle immediately after landing may provide solutions to the randomness of flight time and further increase the CFJS. Figure 3f demonstrates the initial position and four landing points during continuous forward jumping on the PVC plate with a speed of 250.1 mm/s (3.85 body lengths per second). The angle deviation of each jump was less than 8°, which means that the robot can achieve a reasonably good straight-line movement. More details

on the LSJR's motion precision can be seen in the Supplementary Note 2 and Supplementary Movie 8.

**Steered jumping of a dual-body LSJR.** Steered jumping is a useful function for animal locomotion in unstructured and complex terrains. It is desirable for soft jumping robots to replicate this ability. Two LSJRs were connected abreast, resulting in a dual-body LSJR (Fig. 4a, b) with the ability to adjust its locomotion direction[42]. Applying a voltage ($V_1$) to one unit of the LSJRs causes the unit to jump: it deforms, generates forward kinetic energy, and bears ground friction forces. The center of gravity of the dual-body LSJR is not in the direction of the initial speed and friction forces, resulting in turning behavior (each cycle achieves a turning angle of $\alpha$, as shown in Fig. 4a). Selectively controlling $V_1$ on the left and right LSJRs results in steered jumping. To illustrate the turning performance of the dual-body LSJR, it was made to turn at least 60° on each of the four substrates. Figure 4c shows that the robot on the wood plate achieved a greater turning speed (TS). An average TS = 138.4°/s was achieved at 10 kV and 4 Hz, which, to the best of the authors' knowledge, is the fastest among existing soft jumping robots. An average TS of only 27.9°/s was achieved on the glass plate at 10 kV and 4 Hz because the surface was smooth. Figure 4d demonstrates

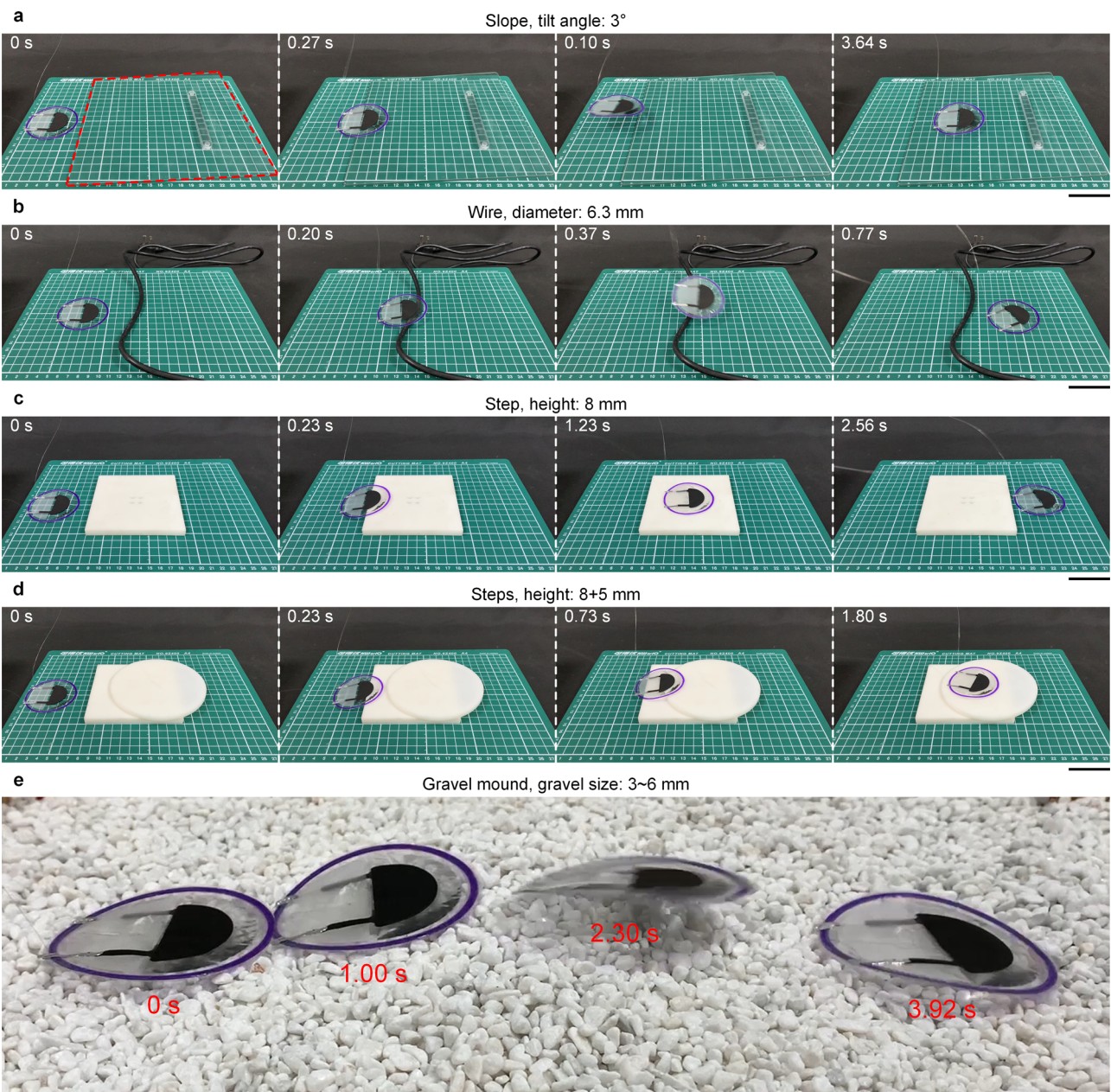

**Fig. 5 Single-unit LSJR obstacle crossing at 10 kV and 4 Hz.** See also Supplementary Movie 5. **a** Climbing on the glass plate (tilt angle of 3°) with a CFJS of 16.3 mm/s. **b** Crossing an electric wire (diameter of 6.3 mm). **c** Jumping across a square step (height of 8 mm). **d** Jumping across continuous steps (heights of 8 and 5 mm). Scale bar, 5 cm. **e** Composite image of the LSJR's locomotion on a gravel mound (gravel size: 3–6 mm). Scale bar, 2 cm.

the initial and final positions during a continuous turning procedure on the PVC plate with a speed of 65.0°/s at 10 kV and 4 Hz. Sufficient substrate surface roughness not only prevents the robot from slipping during continuous motions but also hinders the movement of the unpowered LSJR and thus the turning behavior. Therefore, a potential solution is to apply a large $V_1$ to one LSJR and a small $V_0$ to the other LSJR. This will lead to a greater jump by one unit, thus resulting in turning behavior, and will be considered in our future work. More details on the LSJR's motion precision can be seen in Supplementary Note 2 and Supplementary Movie 8.

**Obstacle-crossing ability of the LSJR.** A main function of a jumping robot is obstacle avoidance so that it can conduct

explorations, inspections, and reconnaissance tasks in complex and unstructured environments. Both the single-body and dual-body LSJRs can be used to achieve decent obstacle-crossing capability, as shown in Figs. 5 and 6 (Supplementary Movies 5 and 6). Under an applied voltage of 10 kV and an actuation frequency of 4 Hz, the single-body LSJR was found to (1) climb on the glass plate (tilt angle of 3°) with a CFJS of 16.3 mm/s (0.25 body lengths per second), (2) jump cross an electric wire (diameter of 6.3 mm), (3) jump across a square step (8 mm high), and (4) jump across continuous steps (consisting of the square step and a 5 mm high round step). In crossing tests with an obstacle height interval of 4 mm, the maximum height that the LSJR can cross was 14 mm for cuboids, and 18 mm for triangular prisms and cylinders, as shown in Supplementary Fig. 11 (Supplementary Movie 7). Affected by the leaping posture and wires, the

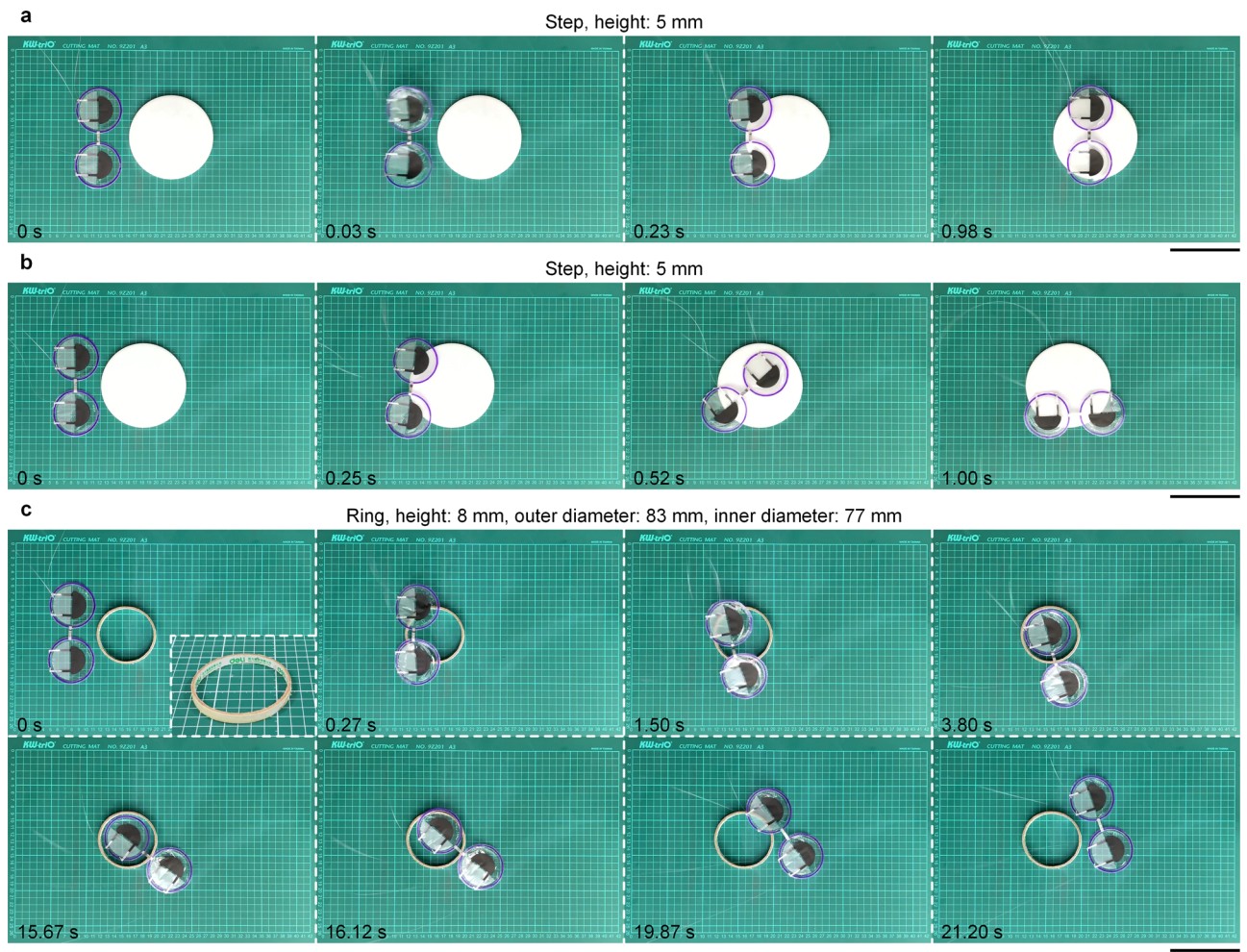

**Fig. 6 Dual-body LSJR obstacle crossing.** See also Supplementary Movie 6. **a** Straight jumping across a round step (height of 5 mm). **b** Steered jumping across the round step (height of 5 mm). **c** Jumping across a ring obstacle (height of 8 mm, inner diameter of 77 mm, and outer diameter of 83 mm). Scale bars, 10 cm.

maximum obstacle height that the LSJR can cross was less than the max JH. In order to further prove the robot's ability to overcome obstacles in realistic unstructured environments, we made a gravel mound with many gravels (size: 3 to 6 mm), and tested LSJR's locomotion on the gravel mound (Fig. 5e). The steered jumping (simultaneous jumping and turning) capability of the dual-body LSJR made it (1) jump across a 5 mm high round step and (2) jump across a ring object with a height of 8 mm, an inner diameter of 77 mm, and an outer diameter of 83 mm. It was challenging for the single-body LSJR to avoid the ring obstacle. The collaborative movement of two LSJR units results in much greater flexibility and improved obstacle-crossing ability for soft jumping robots, thus making the robot more adaptable to complex terrains.

## Discussion

We proposed an electrohydrostatically driven, low-profile (0.85-mm-thick), lightweight (1.1 g), modular, and cost-effective tethered LSJR based on a sEHBA. The robot features capabilities of rapid, continuous, and steered jumping, load-carrying, and obstacle-crossing through a simple control strategy. The inspiration for the design of the sEHBA stemmed from the electrohydrostatic jumping of HASEL actuators[40] and the periodic saddle-shaped bending caused by the predeformed frame of DEAs[42]. In the design process, a mechanical analysis model and a

dielectric liquid's center of gravity moving equivalent model were built to guide the optimization of size parameters.

Most existing soft jumping robots (Table 1) have a large JH but require a long actuation/energy-storage time and righting time, which leads to a slow CFJS and lack of flexibility. A few soft jumping robots[26,37] have a high actuation frequency and high speed mobility, but they can only achieve a small JH (<0.25 body height), which is not suitable for overcoming challenging obstacles. Our LSJR used the special liquid–air layout and the edge-fixing prebending frame to achieve rapid continuous forward and steered jumping locomotion caused by periodic saddle-shaped bending and anisotropic liquid flow, which made up for the limitations of HASEL actuators[40], including (1) unachievable forward and steered jumping, (2) weak single-jump performance without stacking, and (3) incapability for rapid restoration. As a result, the LSJR had a short actuation time (~10 ms) and was able to achieve a JH of 7.68 body heights, a JD of 1.46 body lengths in a single jump (Fig. 2 and Supplementary Movie 2), and a CFJS of 390.5 mm/s (6.01 body lengths per second) with a frequency of 4 Hz (Fig. 3 and Supplementary Movie 3). The angle deviation of each jump can be controlled within 8° in continuous forward jumping locomotion. The integration of two LSJRs was able to readily achieve rapid steered jumping (Fig. 4 and Supplementary Movie 4). The TS of the dual-body LSJR was able to reach 138.4°/s, which is the fastest among existing soft jumping robots. Experiments also verified that LSJR's rapid continuous jumping locomotion could be applied to cross many obstacles,

**Table 1 Comparison between this work and some soft forward jumping robots.**

| Soft jumping robots | Energy-storing jumping | Actuation methods | Weight (g) | Jumping distance | Jumping height | Propulsive interval time (s) | Landing stability | Straight line jumping capability | Direction-adjusting capability | Unit cost |
|---|---|---|---|---|---|---|---|---|---|---|
| Kovač (2013)[7] | Yes | Spring | 14.00 | ≈2.55 BD | ≈3.44 BD | ≥5.00 | Medium | Low | Directional jumping (∞ directions) | High |
| Zhakypov (2019)[18] | Yes | SMA | 9.70 | 3.97 BL | 2.50 BH | >24.00 | High | High | No | High |
| Huang (2018)[19] | Yes | SMA | 3.00 | ≈2.00 BL | 1.00 BH | ≈3.00 | High | High | No | High |
| Hu (2018)[21] | Yes | Magnetic | N/A | ≈1.63 BL | 2.44 BL | >10.00 | Low | Medium | Steered jumping (≈15.0°/s) | N/A |
| Ahn (2019)[22] | Yes | Light | N/A | 8.00 BL | 5.00 BH | ≥100.00 | High | High | No | N/A |
| Hu (2017)[23] | Yes | Light | N/A | N/A | 5.00 BH | ≈10.56 | High | High | No | N/A |
| Duduta (2020)[25] | Yes | DEA | 0.90 | 1.34 BL | 1.16 BL | ≥6.00 | High | High | No | Low |
| Zhao (2019)[26] | No | DEA | 6.50 | ≈0.29 BL | <0.25 BH | ≈0.03 | High | High | No | Low |
| Ni (2015)[27] High | No | Pneumatic | | N/A | ≈0.64 BL | ≈0.64 BH | ≥0.75 | High | High | No |
| Liu (2020)[29] Low | No | Pneumatic | | 0.45 | ≈0.90 BL | ≈0.80 BL | ≥0.28 | High | High | No |
| Tolley (2014)[30] | No | Chemical | 510.00 | 7.50 BH | 7.50 BH | ≥0.03 | Low | Low | Directional jumping (3 directions) | High |
| Loepfe (2015)[31] | No | Chemical | 2100.00 | 2.78 BD | 1.11 BD | ≈4.50 | Medium | Low | No | High |
| Bartlett (2015)[32] | No | Chemical | Tethered: 478.60 Untethered: 964.60 | Tethered: N/A Untethered: 0.50 BL | Tethered: 2.35 m Untethered: 6.00 m | ≥2.45 | High | Low | Directional jumping (3 directions) | High |
| Churaman (2011)[33] | No | Chemical | 0.314 | ≈21.78 BL | 80 mm | ∞ | Low | Low | No | Low |
| Li (2017)[34] | No | Motor | 250.00 | 2.57 BL | 1.00 BH | ≈10.00 | Medium | Low | Steered jumping (≈0.6°/s) | High |
| Mintchev (2018)[36] | No | Motor | 37.00 | ≈3.35 BD | 2.86 BD | ≈3.00 | Medium | Low | No | High |
| Wu (2019)[37] | No | PVDF | ≈0.06 | ≈0.11 BL | <0.25 BH | <0.01 | High | High | Steered jumping (≈0.8°/s) | Low |
| This work | No | sEHBA | 1.10 | 1.46 BL | 7.68 BH | ≈0.01 | High | High | Single-body: No Dual-body: steered jumping (138.4°/s) | Low |

Notes: *BL* body length, *BH* body height, *BD* body diameter, *N/A* not available. Landing stability, straight line jumping, and unit cost were evaluated in three (high, medium, and low) levels. The following are the detailed level judgment criteria:

● Landing stability:
(1) High. After the robot lands, it does not roll and does not need artificial/self-righting before the next jumping.
(2) Medium. The robot is capable of self-righting. After landing, it rolls a distance and sometimes needs self-righting based on its posture.
(3) Low. The robot is not capable of self-righting. After landing, it rolls a distance and sometimes needs artificial-righting based on its posture.

● Straight line jumping capability:
(1) High. There is no need for artificial direction adjustment in continuous forward jumping process. Connecting several continuous landing points as a line, it is basically a straight line.
(2) Medium. The robot needs artificial direction adjustment in continuous forward jumping process. Connecting several continuous landing points as a line, it is basically a straight line.
(3) Low. The landing points have some randomness in continuous jumping process. Connecting several continuous landing points as a line, it is a curve.

● Unit cost:
We estimate the unit price based on the main material and component price (from a Chinese e-commerce website) of the robot.
(1) High. The unit price exceeds 10 RMB.
(2) Low. The unit price does bot exceeds 10 RMB.

including slopes, wires, single steps, continuous steps, ring obstacles, gravel mounds, and cubes of different shapes. The maximum height of the robot can reach up to 18 mm (Figs. 5 and 6, Supplementary Movie 5, Supplementary Fig. 11 and Supplementary Movies 6 and 7).

The jumping performances of the LSJR rely on not only the applied voltages but also the surface textures of various substrates (Supplementary Fig. 10). Under the same applied voltage (10 kV, 4 Hz), the glass substrate with the smoothest surface provided the lowest friction among all substrates, leading to a lower CFJS of 95.6 mm/s (1.47 body lengths per second) and a lower TS of 27.9°/s. This currently limits the application of the robot to jumping on relatively smooth surfaces but can be mitigated if electroadhesion is applied by adding additional passive electrodes to the rear pouch.

The LSJR can be applied to detect and record environmental changes such as temperature and ultra-violet light by attaching a light and soft temperature sensor/paste and photochromic dyes (Supplementary Fig. 15 and Supplementary Movie 9). Through integrating other sensors, it is expected to detect more environmental factors, such as pollutants in industrial environments and civil buildings. In addition, future work may also include (1) study of the scalability and parametric optimization of the sEHBA to achieve better jumping performance, (2) development of an untethered LSJR and its applications, and (3) investigation of sEHBA's other soft robotics applications such as wall-climbing robots, swimming robots, and flapping wing robots.

## Methods

**Materials of the LSJR**. The inextensible pouch shell was made by heat-sealing two 16-μm-thick BOPP films (Jiazhixing Co., China) and injected with air and a dielectric liquid made of 25# mineral transformer oil (Aokelai Lubricants Co., China). BOPP has reasonably good dielectric breakdown strength (~700 V μm$^{-1}$) and tensile strength (~300 N mm$^{-2}$)[44]. The dielectric liquid has favorable dielectric properties and low viscosity[45]. The electrodes were screen printed on the BOPP films using LN-GCI-3 graphene conductive inks (Jining Leadernano Tech., China). The plastic ring frame was fabricated by laser cutting a 0.5-mm-thick PVC sheet (Lizhiyuan Plastic Industry, China), which has good mechanical properties and low density[46]. All materials are low in cost and easy to procure.

**Fabrication procedure of the LSJR**. The fabrication procedure of an LSJR involves nine major steps (Supplementary Fig. 4). First, we screen printed conductive inks onto BOPP films using a screen-printing machine (Mingming Screen-Printing Equipment Co., China) (Supplementary Fig. 5). The electrodes were cured at room temperature for 12 h, resulting in a flexible graphene electrode with a thickness of 20 μm. Second, we stacked two BOPP-electrode composite films with the electrodes facing outwards and put them on a Teflon high-temperature cloth as a load-dispersing layer. Third, the soldering iron was set to 200 °C to heat-seal the BOPP films, creating two semicircular pouches (both 55 mm in diameter) and leaving two fill ports in the seal of each pouch. Fourth, we filled the front pouch with the dielectric liquid of 1 mL and the rear pouch with the same volume of air by using two syringes. Fifth, we squeezed the air bubbles out of the front pouch and heat-sealed the two fill ports to prevent fluid compression. Sixth, we lifted the four corners of the top BOPP film in sequence and placed the laser-cut flexible PVC ring frame (thickness of 0.5 mm, inner diameter of 58 mm, and outer diameter of 62 mm) between the two BOPP films. Seventh, we fixed the edge position of the three-layer composite membrane and moved the soldering iron on the films to deform the ring frame and heat-seal it following the rebound trajectory. In this step, the prebending levels of the final LSJR can be controlled by moving different distances ($d$). Eighth, we heat-sealed along the inside and outside circles of the predeformed frame and removed the pushpins to rebound the frame to generate prebending. Finally, excess BOPP film was cut away, and two aluminum tapes were attached to the ends of electrodes to create a completed LSJR with a total body weight of 1.1 g and a full body length of 65 mm.

**Experimental control strategy of the LSJR**. After electrical connections were made to the LSJR, applying a high voltage to the LSJR for 10 ms enabled it to complete a jump. In the single-jump tests, after it took-off, the aluminum pieces at the tail detached from the copper electrodes fixed on the ground, thereby cutting off the voltage. The voltage waveform was a square wave and the duty cycle was 1:6001, as shown in the top right inset of Fig. 2a. Voltage reversal and a wait time of 60 s were conducted to alleviate the residual charge issue. Different voltages caused different jumping distances and JHs. In the continuous locomotion tests, the wires

were used for electrical connection instead of using a separable aluminum–copper contact structure. The voltage waveform was a square wave and the duty cycle was adjustable, as shown in Fig. 3e. The power-on time of the robot in each cycle was 10 ms. Different voltages and frequencies caused different continuous forward jumping speeds. In the direction-adjusting tests, two LSJRs were abreast connected and the voltage was selectively applied to one of them. The electrical connections were made the same as in the continuous locomotion tests. The voltage waveform was a square wave and the frequency was 4 Hz. The power-on time of the robot in each cycle was 10 ms. Different voltages caused different turning speeds. In the obstacle-crossing tests, the electrical connections were the same as those in the continuous locomotion tests and direction-adjusting tests. The voltage waveform was a square wave and the frequency was 4 Hz. The voltage was 10 kV and the power-on time of the robot in each cycle is 10 ms.

**Experimental set up of the LSJR**. For single-jump tests, a high-speed camera (2F01, Revealer, China) was used to capture the motions. For other low-speed tests, a Nikon D3400 DSLR camera and an Apple iPhone XR were used to record the movements. For substrate surface morphology characterization, a 3D Laser Scanning Confocal Microscope (OLS4000, Olympus Corp., Japan) was used to measure the areal surface texture. Adobe Premiere (version 7.0.0) and Adobe Photoshop (version 19.1.4) were used to process the videos and to obtain the experimental data. An EMCO high-voltage amplifier (E101CT, EMCO High Voltage Co., USA) was used with two high-voltage relays (CRSTHV-14KV-A, CRST, China), controlled by a DMAVR-128 board (Ningbo Xinchuang Electronic Technology Co., China), to energize the LSJR.

## Data availability

The authors declare that data supporting the findings of this study are available within the paper and its Supplementary Information files.

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

## Acknowledgements

This work was supported by the National Key Research and Development Project of China (grant no. 2020YFB1313000) and the National Natural Science Foundation of China (grant nos. 52075051 and 51975070). J.G. wants to thank Prof. Jonathan Rossiter for supporting this work whilst working at SoftLab Bristol, University of Bristol, under grant EP/M020460/1. In addition, he wants to thank Prof. Jinsong Leng for the support while working at HITSZ. We also thank Yi Sun and Thomas Bamber for the initial review of the paper, and Wenbo Liu for the assistance in experiments.

## Author contributions

R.C. conceived the research. R.C. and Z.Y. jointly designed and implemented the soft electrohydrostatic bending actuator. J.G. refined the legless soft jumping robot concept, and proposed the robot design rationale and refined paper novelty clarifications. Z.Y. and X.Z. conducted the structural design, device manufacturing and experiments. Z.Y., R.C., L.B. and X.Z. analyzed and interpreted the results. Z.Y. drafted the manuscript. Z.Y., X.Z. and J.G. designed and optimized the figures, tables, and videos. Z.Y., J.G., L.W. and Y.S. wrote the abstract, introduction, and discussion part, and fully revised the whole manuscript. L.B., F.L., H.P., L.X., Y.P. and J.L. reviewed and commented on the paper.

## Competing interests

The authors declare no competing interests.
