## [Peer Review File · Nature Communications]

REVIEWER COMMENTS

Reviewer #1 (Remarks to the Author):

The paper deals with the design, realization, and testing of a novel legless soft jumping robot. The proposed jumping robot is electrohydrostatically driven by an original actuation concept. Robot jumping performances are at the top of the state of art.

The addressed topic is hot for the robotics research community and the proposed work proposes novel and interesting ideas and viewpoints.

The paper is clear, well-written, and easy to follow. It provides a fair level of details for all the parts of the work (concept, design, fabrication, performance evaluation).

The reviewer recommends it for publication.

The reviewer has a couple of comments. The first is about the "obstacle crossing ability". It would be interesting to perform this part of the work in a bit more rigorous way, for instance by parametrizing the obstacle shape (e.g., rectangle, triangle, spherical, ...) and to know the size of the obstacle that can be overcome by the robot. The reviewer expects that in general, the max obstacle height is less than the max jumping height and that the max obstacle height depends on the robot motion and obstacle shape. The reviewer thinks this would add additional value to the paper.

The second comment is related to robot control. Given the explorative nature of the proposed work the control is not addressed in the paper, however, it would be extremely interesting to quantify the precision with which the translational and rotational motion can be achieved to have (even a rough) idea of the error with which the robot is able to reach a given (even discrete) position and orientation on the ground.

Reviewer #2 (Remarks to the Author):

The paper presents an interesting technological development. The resulting jumping robot represents an interesting achievement, in terms of technical solution for actuation. The principle used for obtaining a jumping behaviour is different from those used in other robots in literature.

The paper is well written but some claims and conclusions sound a bit overstated. The jumping behaviour, especially at high frequencies, is more similar to vibration-based locomotion. Flight

phases are not so high and long, even if resulting numbers are high, in terms of body height, given the flat shape of the robot.

The main question is to what extent the robot locomotion can be controlled and programmed. It is not clear what tasks the robot can perform.

The technical details are provided thoroughly.

Response to the Decision Letter

Dear editors and reviewers,

We are grateful for your time evaluating our paper, entitled “Legless soft robots capable of rapid, continuous, and steered jumping” (Manuscript Number: NCOMMS-21-23410). We appreciate your thorough and constructive comments and suggestions, which have helped us greatly in improving our paper.

In the revised manuscript, we have tested the size of the obstacles that the LSJR can overcome and evaluated the precision of the translational and rotational motion that can be achieved. We have revised the description of some claims and conclusions which may sound overstated. We have changed expressions in the manuscript to make them more rigorous. We have answered the reviewer’s question about the jumping behavior and to what extent the LSJR locomotion can be controlled and programmed. We have added an experiment to show the dual-body LSJR can move along a given route. Potential applications the robot were also added.

The contents of the paper have been revised accordingly. Please see the detailed responses to your comments below.

The revised figures:

- The new Figure S11 has been added to show the size of the obstacles of cuboids, triangular prisms and cylinders, which the LSJR can overcome.
- The new Figure S12 has been added to show the trajectories of continuous forward jumping of the single-unit LSJR.
- The new Figure S13 has been added to show the trajectories of turning of the dual-body LSJR.
- The new Figure S14 has been added to show the trajectories of alternate moving of the dual-body LSJR.
- The new Figure S15 has been added to show the applications of the LSJR, including detection of temperature and ultra-violet light.

The revised movies:

- The new Movie S7 has been added to show the crossing tests of the large, medium and small obstacles of cuboids, triangular prisms and cylinders.
- The new Movie S8 has been added to show the trajectories of the LSJR in different motion modes.

- The new Movie S9 has been added to show the applications of the LSJR, such as the detection of temperature and ultra-violet detection.

Best regards,

Prof. Rui Chen

Email: cr@cqu.edu.cn

Chongqing University

Prof. Huayan Pu

Email: phygood_2001@shu.edu.cn

Shanghai University

Prof. Jianglong Guo

Harbin Institute of Technology (Shenzhen)

Prof. Li Wen

Beihang University

Prof. Yu Sun

University of Toronto

Response to Referee 1:

Comment 1:

The first is about the "obstacle crossing ability". It would be interesting to perform this part of the work in a bit more rigorous way, for instance by parametrizing the obstacle shape (e.g., rectangle, triangle, spherical, ...) and to know the size of the obstacle that can be overcome by the robot. The reviewer expects that in general, the max obstacle height is less than the max jumping height and that the max obstacle height depends on the robot motion and obstacle shape. The reviewer thinks this would add additional value to the paper.

Response 1:

Thank you for the valuable comments. We have added experiments for the LSJR to cross the obstacles of cuboids, triangular prisms, and cylinders. We also tested the maximum obstacle size that the robot can overcome in the crossing tests (height increment: 4 mm). The maximum height that the LSJR can cross is 14 mm for cuboids, and 18 mm for triangular prisms and cylinders. The robot's center of gravity is in the front half, causing it to turn into a diving posture with the head falling first, which is not conducive to jumping over higher obstacles. Therefore, the actual maximum obstacle height is less than the maximum jumping height due to the limits of the motion posture and wires.

Revised text:

Furthermore, we show that LSJR's rapid continuous jumping locomotion can cross various obstacles, including slopes, wires, single steps, continuous steps, ring obstacles, gravel mounds, and cubes of different shapes, some of which are larger than the robot. (Page 2, Line 86)

In crossing tests with an obstacle height interval of 4 mm, the maximum height that the LSJR can cross was 14 mm for cuboids, and 18 mm for triangular prisms and cylinders, as shown in Supplementary Fig. 11 (Supplementary Movie 7). Affected by the leaping posture and wires, the maximum obstacle height that the LSJR can cross was less than the max jumping height. (Page 10, Line 321)

Experiments also verified that LSJR's rapid continuous jumping locomotion could be applied to cross many obstacles, including slopes, wires, single steps, continuous steps, ring obstacles, gravel mounds, and cubes of different shapes. The maximum height of the robot can reach up to 18 mm (Fig. 5, Fig. 6, Supplementary Movie 5, Supplementary Fig. 11, Supplementary Movie 6, and Supplementary Movie 7). (Page 14, Line 381)

Supplementary Fig. 11. Single-unit LSJR crossing for obstacles of different shapes and sizes. See Supplementary Movie 7. **(a)** Crossing tests for three cuboids (height of 10 mm, 14mm and 18 mm). The LSJR collided with the front wall of the 18 mm height cuboid. **(b)** Crossing tests for three triangular prisms (height of 14 mm, 18mm and 22 mm). The LSJR slipped down from the 22 mm height triangular prism. **(c)** Crossing tests for three cylinders (height of 14 mm, 18mm and 22 mm). The LSJR slipped down from the 22 mm height cylinder. Scale bar, 4 cm. (Page 36, Line 825)

Comment 2:

The second comment is related to robot control. Given the explorative nature of the proposed work the control is not addressed in the paper, however, it would be extremely interesting to quantify the precision with which the translational and rotational motion can be achieved to have (even a rough) idea of the error with which the robot is able to reach a given (even discrete) position and orientation on the ground.

Response 2:

According to your suggestions, we have tested the precision of translational and rotational motion of the LSJR. Under the applied voltage of 4 Hz and 10 kV, we recorded five trajectories of continuous forward jumping of a single-unit LSJR and five trajectories of tuning of a dual-body LSJR. In the continuous forward jumping test of 37.5 cm, the single-unit LSJR showed an average of 0.62 lateral deviation (maximum 1.2 cm) and 0.95° angle deviation (maximum 1.67°). In the turning test of 90°, the dual-body LSJR showed an average of 3.13° angle deviation (maximum 5.13°). To move along a given route, the dual-body LSJR can be used to imitate the movement of bipedal animals by alternately activating one of the two units in the robot. Continuous turning can adjust the positions of the robot in the process of moving along the predetermined route.

Revised text:

More details on the LSJR's motion precision can be seen in the Supplementary Note 1. (Page 8, Line 277)

Under the applied voltage of 4 Hz and 10 kV, the single-unit LSJR can be used to conduct a 37.5 cm long translational motion by continuous forward jumping. Five trajectories of continuous jumps are shown in Supplementary Fig. 12a. When the robot reached the end, the lateral deviation averaged 0.62 cm (maximum 1.2 cm), and the angle deviation averaged 0.95° (maximum 1.67°). The whole jumping process can be seen in Supplementary Fig. 12b and Supplementary Movie 7. (Page 25, Line 749)

Under the applied voltage of 4 Hz and 10 kV, the dual-body LSJR completed a 90° turn by continuously actuating one unit of the dual-body LSJR. Five trajectories of turns are shown in Supplementary Fig. 13a. When the robot turned 90° after five jumps, the angle deviation averaged 3.13° (maximum 5.13°). The whole turning process can be seen in Supplementary Fig. 13b and Supplementary Movie 8. (Page 25, Line 756)

To move along a given route, the dual-body LSJR imitates the movement of bipedal animals, by alternately activating one of the two units in the robot. Continuous steering can adjust the positions to move on the predetermined route. The whole moving process can be seen in Supplementary Fig. 14 and Supplementary Movie 9. The dual-body LSJR can jump forward, turn or walk like two feet to reach the given position and orientation. Smaller displacements can be achieved by lowering the applied voltage. (Page 25, Line 762)

Supplementary Fig. 12. Continuous forward jumping trajectories of the single-unit LSJR. See Supplementary Movie 8. **(a)** Trajectories of five continuous forward jumps on the plate at 4 Hz and 10 kV. **(b)** Composite image of the initial position and seven landing points in continuous forward jumping of the yellow trajectory in the line chart. Scale bar, 4 cm. (Page 37, Line 833)

Supplementary Fig. 13. Turning trajectories of the dual-body LSJR. See Supplementary Movie 8. **(a)** Trajectories of five turning on the plate at 4 Hz and 10 kV. Turning angle close to 90° required 5 jumps to achieve. **(b)** Composite image of the initial position and four landing points in continuous forward jumping of the yellow trajectory in the line chart. Scale bar, 4 cm. (Page 38, Line 839)

Supplementary Fig. 14. Alternate moving trajectories of the dual-body LSJR. See Supplementary Movie 8. The dual-body LSJR corrected the positions in its forward process by steering to close to the predetermined straight line at 10 kV. Scale bar, 4 cm. (Page 39, Line 845)

Response to Referee 2:

Comment 1:

The paper is well written but some claims and conclusions sound a bit overstated. The jumping behaviour, especially at high frequencies, is more similar to vibration-based locomotion. Flight phases are not so high and long, even if resulting numbers are high, in terms of body height, given the flat shape of the robot.

Response 2:

- 1) Thank you for your encouraging remarks and instructive comments. We have revised expressions in the manuscript to make them more rigorous.
- 2) In the experiments, we have tested the locomotion performance of the LSJR at different frequencies (from 0.5 Hz to 8 Hz) to obtain the best actuating frequency. At the applied voltage of high frequencies, the jumping behaviour is indeed similar to vibration-based locomotion. This is because the robot is still in the air when another actuation cycle started, causing the next jumping performance to be impaired.
- 3) Compared with the cockroach robot²² and the hopping-running robot²⁶, the LSJR is capable of crossing higher and more complex obstacles. Meanwhile, compared with the dielectric elastomer robots²⁴⁻²⁵, it demonstrates that the efficiency of obstacle-crossing for robots not only depends on the jumping height, but the jumping frequency is also important.

Comment 2:

The main question is to what extent the robot locomotion can be controlled and programmed. It is not clear what tasks the robot can perform.

Response 2:

- 1) The single-unit LSJR can be controlled to jump forward continuously to achieve straight moving. The dual-body LSJR can achieve clockwise or counterclockwise turning by controlling one of the two units. According to your suggestions, we have designed experiments to test the precision of straight moving and turning of the LSJR. After the single-unit LSJR moved about 37.5 cm in a straight line, it generated an average of 0.62 lateral deviation (maximum 1.2 cm) and 0.95 angle deviation (maximum 1.67°). After the dual-body LSJR turned about 90°, it generated an average of 3.13° angle deviation (maximum 5.13°). Additionally, the dual-body LSJR can adjust its position in the process of moving by turning many times to along a predetermined route as walking on two feet. More precise positions can be achieved by reducing the applied voltage.
- 2) Due to its lightweight, compactness, flexibility, and adaptability, the LSJR can enter narrow environments for detection. We integrated light and flexible sensors on the robot to detect environmental factors, such as temperature and ultra-violet light. The results show that the LSJR with a temperature sensor/paste moved in a 55°C environment, and the color of the temperature paste changed from white to black after about 30 s. For the detection of ultraviolet light, we printed photochromic dye on the LSJR. The color changed from white to red within 2 seconds after entering the area with ultra-violet light. Both of the above two color-changing materials can maintain the changed color for several minutes, and the

recorded signal has enough time to be collected.

Revised text:

More details on the LSJR's motion precision can be seen in the Supplementary Note 1. (Page 8, Line 278)

The LSJR can be applied to detect and record environmental changes such as temperature and ultra-violet light by attaching a light and soft temperature sensor/paste and photochromic dyes (Supplementary Fig. 15 and Supplementary Movie 9). Through integrating other sensors, it is expected to detect more environmental factors, such as pollutants in industrial environments and civil buildings. (Page 14, Line 395)

Under the applied voltage of 4 Hz and 10 kV, the single-unit LSJR can be used to conduct a 37.5 cm long translational motion by continuous forward jumping. Five trajectories of continuous jumps are shown in Supplementary Fig. 12a. When the robot reached the end, the lateral deviation averaged 0.62 cm (maximum 1.2 cm), and the angle deviation averaged 0.95° (maximum 1.67°). The whole jumping process can be seen in Supplementary Fig. 12b and Supplementary Movie 7. (Page 25, Line 749)

Under the applied voltage of 4 Hz and 10 kV, the dual-body LSJR completed a 90° turn by continuously actuating one unit of the dual-body LSJR. Five trajectories of turns are shown in Supplementary Fig. 13a. When the robot turned 90° after five jumps, the angle deviation averaged 3.13° (maximum 5.13°). The whole turning process can be seen in Supplementary Fig. 13b and Supplementary Movie 8. (Page 25, Line 753)

To move along a given route, the dual-body LSJR imitates the movement of bipedal animals, by alternately activating one of the two units in the robot. Continuous steering can adjust the positions to move on the predetermined route. The whole moving process can be seen in Supplementary Fig. 14 and Supplementary Movie 9. The dual-body LSJR can jump forward, turn or walk like two feet to reach the given position and orientation. Smaller displacements can be achieved by lowering the applied voltage. (Page 25, Line 762)

Supplementary Fig. 12. Continuous forward jumping trajectories of the single-unit LSJR. See Supplementary Movie 8. **(a)** Trajectories of five continuous forward jumps on the plate at 4 Hz and 10 kV. **(b)** Composite image of the initial position and seven landing points in continuous forward jumping of the yellow trajectory in the line chart. Scale bar, 4 cm. (Page 37, Line 829)

Supplementary Fig. 13. Turning trajectories of the dual-body LSJR. See Supplementary Movie 8. **(a)** Trajectories of five turning on the plate at 4 Hz and 10 kV. Turning angle close to 90° required 5 jumps to achieve. **(b)** Composite image of the initial position and four landing points in continuous forward jumping of the yellow trajectory in the line chart. Scale bar, 4 cm. (Page 38, Line 835)

Supplementary Fig. 14. Alternate moving trajectories of the dual-body LSJR. See Supplementary Movie 8. The dual-body LSJR corrected the positions in its forward process by steering to close to the predetermined straight line at 10 kV. Scale bar, 4 cm. (Page 39, Line 841)

Supplementary Fig. 15. Applications of the LSJR. See Supplementary Movie 9. **(a)** Temperature detection. The color of temperature paste (40□) changed from white to black when heated on the 55□ platform. **(b)** Ultra-violet light detection. The color of the photochromic dye changed from white to red when exposed to ultra-violet light. Scale bar, 4 cm. (Page 40, Line 846)

REVIEWERS' COMMENTS

Reviewer #1 (Remarks to the Author):

The authors successfully address the reviewer's comments

Reviewer #2 (Remarks to the Author):

The authors added helpful information to the paper, in response to the comments given in the review. Few aspects have been clarified. Results are better described and illustrated by pictures and videos.

Response to the Decision Letter

Dear editors and reviewers,

We are grateful for your time evaluating our paper, entitled “Legless soft robots capable of rapid, continuous, and steered jumping” (Manuscript Number: **NCOMMS-21-23410B**). We appreciate your thorough and constructive comments and suggestions, which have helped us greatly in improving our paper.

In the revised manuscript, we have revised the abstract as you proposed and the information of authors. We have deleted the words such as new/novel/first. We have revised the description of some claims and conclusions. We have checked all the guidance listed in the author checklist and given the response to the editorial requests. We have responded the reviewers’ comments.

The contents of the paper have been revised accordingly. Please see the detailed responses to your comments below.

The revised figures:

- The Figure S15 has been revised to describe the ultra-violet light detection rigorously.

The revised movies:

- The Movie S6 has been revised to describe the obstacle crossing experiments of dual-body LSJR rigorously.

Best regards,

Prof. Rui Chen

Email: cr@cqu.edu.cn

Chongqing University

Prof. Huayan Pu

Email: phygood_2001@shu.edu.cn

Shanghai University

Prof. Jianglong Guo

Harbin Institute of Technology (Shenzhen)

Prof. Li Wen

Beihang University

Prof. Yu Sun

University of Toronto

Response to Referee 1:

Comment 1:

The authors successfully address the reviewer's comments.

Response 1:

We are excited for your concurrence to our revision. Thanks again for your thorough and constructive comments and suggestions.

Response to Referee 2:

Comment 1:

The authors added helpful information to the paper, in response to the comments given in the review. Few aspects have been clarified. Results are better described and illustrated by pictures and videos.

Response 1:

Thanks again for your thorough and constructive comments and suggestions in the previous round of review.